# Research on the Impact of an AI Voice Assistant’s Gender and Self-Disclosure Strategies on User Self-Disclosure in Chinese Postpartum Follow-Up Phone Calls

**DOI:** 10.3390/bs15020184

**Published:** 2025-02-10

**Authors:** Xinxin Sun, Tianyuan Shen, Qianling Jiang, Bin Jiang

**Affiliations:** 1School of Design Art and Media, Nanjing University of Science and Technology, Nanjing 210094, China; sunxinxin@mail.njust.edu.cn (X.S.); 123109223220@njust.edu.cn (T.S.); 2Faculty of Innovation and Design, City University of Macau, Macau 999074, China; jiangqianling@jiangnan.edu.cn

**Keywords:** AI voice assistant, postpartum follow-up phone call, self-disclosure, stereotypes

## Abstract

This study examines the application of AI voice assistants in Chinese postpartum follow-up phone calls, with particular focus on how interaction design strategies influence users’ self-disclosure intention. A 2 (voice gender: female/male) × 3 (self-disclosure strategies: normal conversation without additional disclosure/objective factual disclosure/emotional and opinion-based disclosure) mixed experimental design (n = 395) was conducted to analyze how the gender and self-disclosure strategies of voice assistants affect users’ stereotypes (perceived warmth and competence), and how these stereotypes, mediated by privacy calculus dimensions (perceived risks and perceived benefits), influence self-disclosure intention. The experiment measured various indicators using a 7-point Likert scale and performed data analysis through analysis of variance (ANOVA) and structural equation modeling (SEM). The results demonstrate that female voice assistants significantly enhance users’ perceived warmth and competence, while emotional self-disclosure strategies significantly improve perceived warmth. Stereotypes about the voice assistant positively affect users’ self-disclosure intention through the mediating effects of perceived risk and benefit, with perceived benefit exerting a stronger effect than perceived risk. These findings provide valuable insights for the design and application of AI voice assistants in healthcare, offering actionable guidance for enhancing user interaction and promoting self-disclosure in medical contexts.

## 1. Introduction

The emergence of Artificial Intelligence (AI) voice assistants has progressively liberated individuals from tedious and complex tasks. One of the most crucial areas for AI voice assistant applications is the healthcare sector. The demanding workload, specialized knowledge requirements, and the service-oriented nature of healthcare environments make the integration of AI to replace certain manual tasks an inevitable trend ([46]). A growing body of research has begun to examine the communication strategies and effectiveness of AI voice assistants in medical guidance, disease diagnosis, and psychological counseling, and significant progress has been achieved ([74]; [54]). In contemporary healthcare systems, follow-up phone calls have become an essential component of post-care services ([13]). This modality is particularly suitable for AI voice assistants, especially in postpartum follow-up, which is both a frequent and stable occurrence in hospitals, especially in China ([78]). Implementing AI voice assistants in this context can significantly enhance hospitals’ operational efficiency while reducing costs ([46]). Additionally, hospitals seek to acquire insights into patients’ authentic experiences during the medical process through follow-up calls. They need to extract reliable and in-depth information to improve service quality, identify issues in healthcare delivery, and refine management processes ([20]). Therefore, encouraging users to disclose personal information more honestly and comprehensively when using AI voice assistants is crucial for hospitals. A study conducted in the United States ([29]) involved 528 pregnant women and utilized an AI-powered voice assistant to predict pregnancy-related risks and engage in conversations with the participants. Compared to a control group, which listed risks and suggested that participants discuss them with their clinicians, the use of the AI voice assistant led to a 16% increase in the rate of actions taken to reduce those risks. In China, a study ([64]) designed a chat framework for AI-driven health assistants within health communities. It reduced participants’ negative emotions following AI-generated replies, with a 36% decrease. The prevalence of positive emotions was found to be 15% higher in participants who received AI-generated replies than in participants who conversed with human operators. Furthermore, Xunfei Healthcare Technology has implemented proactive patient management through AI-driven follow-up phone calls and other methods. The platform was first launched in Anhui and Shandong, covering over 30 major diseases across various departments and reaching 85% of discharged patients. Because of this management, patient compliance has significantly increased ([62]).

Follow-up phone calls have been identified as a beneficial intervention for patients transitioning from the hospital to home ([71]). During these calls, hospital staff can offer care recommendations, improve communication regarding information exchange, assist in managing symptoms and complications, and help ease the transition from hospital to home ([69]). The telephone is an ideal tool for using a set of structured questions to assess clearly defined outcomes, and it is more readily accepted by women ([61]). Additionally, follow-up phone calls represent a cost-effective approach to improving patient satisfaction, enhancing health outcomes, and reducing readmission rates ([69]). In China, hospital staff typically conduct postpartum follow-up phone calls within 1–2 weeks following maternal discharge. These follow-ups begin with inquiries about the health of both the mother and infant, addressing any concerns the mother may have. They continue with surveys on aspects such as medical service quality, nursing care, post-care support, and medical ethics. The information gathered is then reported back to the hospital ([78]). For hospitals, besides ensuring medical standards, service quality often becomes a decisive factor in competitive success ([73]). This feedback mechanism not only supports improvements in healthcare service quality but also contributes to the optimization of hospital management. Sufficient user feedback can help hospitals respond promptly to patients’ health conditions and offer recommendations for follow-up services. Furthermore, the ongoing improvement of medical quality, nursing quality, logistics support, and medical ethics relies heavily on user feedback. A good number of healthcare institutions have made notable progress in management and service optimization through follow-up phone call programs ([20]).

The personalization paradox suggests that in order to receive personalized services that align with their needs, users must provide detailed personal information, often including sensitive data ([9]). However, obtaining truthful user information in the healthcare sector presents challenges, particularly when dealing with sensitive information. Numerous patients may be reluctant to make a completely honest disclosure for a variety of reasons. Studies indicate that patients often withhold details because of fear of self-disclosure or a desire to manage impressions when interacting with healthcare providers ([41]; [51]). Fear of self-disclosure refers to concerns about revealing private, sensitive, or stigmatized information. Impression management involves patients intentionally concealing or altering information to present themselves positively to healthcare professionals ([41]). Lucas ([41]) argues that AI voice assistants have a unique advantage in reducing both self-disclosure anxiety and impression management behaviors. The anonymity afforded by AI voice assistants allows patients to express themselves more freely, enabling hospitals to collect more honest and detailed feedback. Moreover, the non-contact nature of AI voice assistant interactions addresses the needs of users who prefer to avoid public device contact while still receiving convenient and high-quality service ([34]; [28]).

The CASAs (Computers as Social Actors) framework offers valuable insights into the design strategies for AI voice assistants. This framework, which is central to understanding human disclosure behaviors toward AI, posits that when people interact with intelligent agents possessing human-like features, people adopt disclosure strategies similar to those used in human-to-human interaction ([47]). This theory has been widely applied in business, healthcare, online communities, and other domains. Based on the CASA framework, research on human communication can be extended to human–AI interactions and subsequently verified ([47]). In particular, voice characteristics, derived from stereotype theory, are considered to be a critical factor influencing disclosure intention. Studies have also shown that the self-disclosure of one part during communication affects the disclosure behavior of the other. This reciprocal disclosure effect has been demonstrated in interactions between humans and intelligent agents ([53]; [47]). This study asserts that investigating voice characteristics and disclosure strategies can provide guidance for designing effective AI voice assistants.

The application of AI technology in follow-up phone calls in hospital has shown immense potential, not only alleviating the burden on medical workers but also offering users a more flexible and secure mode of interaction. This provides a viable method for enhancing users’ trust in hospitals and encouraging greater willingness to disclose information. In order to make better use of AI voice assistants for postpartum phone calls, further research is needed to optimize interaction between AI and users. This study aims to explore how different AI voice genders and self-disclosure strategies affect users’ perceptions of AI voice assistants, thereby influencing their intention to self-disclose accurately.

## 2. Related Research and Hypotheses

### 2.1. Self-Disclosure: Privacy Calculus Theory

Self-disclosure refers to the process through which individuals reveal personal experiences or information about themselves during interpersonal communication ([43]). Theories that examine the reasons behind self-disclosure behavior include privacy calculus theory, communication privacy management theory, and trust theory. Among these, privacy calculus theory has garnered significant attention. Laufer and Wolfe argued that the calculus of behavior is an essential prerequisite for personal information disclosure ([37]). This theory enables the calculation and measurement of self-disclosure intention in research. Although it has certain limitations, such as overlooking the possibility of users withdrawing midway because of self-protection, the theory is widely recognized and utilized.

Privacy calculus theory was first introduced by Culnan in 1999 ([12]). The theory suggests that users assess and weigh risks and benefits before taking further action. When disclosing private information, users calculate the risks and benefits associated with the disclosure. When the perceived risks are much lower than the benefits, or when the risks of disclosing information are negligible, users are more likely to disclose ([32]). Under the CASA framework, it is proposed that users adopt similar disclosure strategies when interacting with chatbots as they would with humans ([26]). This implies that privacy calculus theory also applies to situations involving users’ interactions with an AI voice assistant. Many studies have shown the applicability of this theory in human–AI dialogues. For example, Ho’s experimental work ([26]) confirmed that the CASA theory applies to self-disclosure in interactions between humans and an AI voice assistant. Augustin ([3]) explored the application of privacy calculus theory in corporate-sponsored AI voice assistants, examining the influence of corporate’s impression on user trust during self-disclosure. [75] ([75]) used privacy calculus theory in online health communities. He found that privacy concerns, information support, and emotional support significantly influence users’ intention to disclose. The broad applicability of privacy calculus theory in these studies suggests its relevance in postpartum follow-up phone call scenarios.

Privacy calculus theory includes two dimensions: perceived risks and perceived benefits. The perceived risks of self-disclosure are influenced by the personal information disclosed during communication, the importance of the information, the individual’s perception of its privacy, the likelihood of exploitation, the severity of consequences, and the individual’s control over how the information is used ([34]). The perceived risk in privacy calculus theory is not necessarily material or tangible, but can also be social or psychological in nature. As mentioned above, common concerns in healthcare services, such as fear of self-exposure and impression management, stem from the perceived impact of disclosed information on one’s social image ([41]). These perceptions of risks have a negative impact on disclosure intention. [72] ([72]) empirically examined the relationship between perceived risks and information disclosure, concluding that perceived risk negatively influences users’ willingness to disclose. [75] ([75]) explored how perceived risks affect disclosure intention in healthcare settings and discussed the negative effects of perceived risk using protection motivation theory (PMT). Vogel ([63]) identified that, in psychological healthcare, individual risk expectations, including fears of being ignored, criticized, or misunderstood, as well as feeling embarrassed when disclosing to strangers, lead to avoidance of necessary self-disclosure during medical encounters. Based on these findings, we posit the following hypotheses:

**H1:** 
*In the scenario of postpartum follow-ups phone calls, perceived risks have a negative effect on disclosure intention.*


The perceived benefits of self-disclosure are reflected in the fact that disclosing information can generate various forms of social support, resulting in positive outcomes for individuals ([27]). Social support can be categorized as emotional support, material support, informational support, or evaluative support. Disclosing personal information provides important data that can facilitate these types of support. The expectation of receiving these forms of support in exchange for disclosing information constitutes the perceived benefits of self-disclosure. In healthcare contexts, users typically exchange information about their physical condition and experiences to acquire medical resources and information from medical workers. During medical follow-ups phone calls, users also seek emotional, material, and evaluative support. Users may hope that sharing personal information will lead the follow-up personnel to empathize with them and provide more psychological support. They may also expect help with their difficulties, or even material compensation, in exchange for their information. When users perceive that the benefits of disclosure are sufficient, their intention to disclose increases. Handler ([24]) found that, in elderly caregiving contexts, users were more willing to disclose personal information to receive medication recommendations to avoid adverse events. [75]’s ([75]) research indicated that informational and emotional support positively influenced users’ disclosure intention. Individuals are willing to disclose their information for support. [66] ([66]) found that personalized services in online health communities enhanced users’ perception of benefits and alleviated privacy concerns. Based on these studies, the following hypothesis was proposed:

**H2:** 
*In the scenario of postpartum follow-up phone calls, perceived benefits have a positive effect on disclosure intention.*


### 2.2. Stereotype Content Model

The stereotype content model (SCM) is one of the most influential theoretical frameworks in interpersonal interactions and judgments. It highlights how humans use two dimensions—perceived warmth and competence—to interpret others’ behaviors in social interactions ([18]). The core idea of this model is that when humans encounter a new social target, they quickly assess the target’s intentions (whether they are a friend or foe) and abilities (whether the target can pose a threat), leading to judgments about the social group the target belongs to. Perceived warmth refers to the positivity of one’s perceived intentions, such as friendliness, helpfulness, and trustworthiness, while competence reflects the ability to perform those intentions ([17]; [70]). This psychological mechanism, rooted in survival and reproduction needs, influences interpersonal interactions and has been extended to human–AI interactions in modern technological contexts ([47]). Jin and Youn’s study ([31]) found that human trust in and attitudes toward AI voice assistants depends on their perceptions of the agents’ warmth and competence. Furthermore, gender stereotypes are particularly relevant in intelligent voice assistants, where the alignment of social gender roles often influences users’ attitudes and behavior toward robots. Lass’s study ([36]) further highlighted the existence of “voice stereotypes”, where people infer characteristics such as gender, height, and weight from voice features. This phenomenon is especially important when it comes to designing AI voice assistants. In this study, stereotype theory was introduced to explore users’ stereotypes of AI voice assistants for postpartum follow-up phone calls.

The formation of stereotypes and privacy calculus are key processes in communication. While stereotype theory influences initial cognitive perceptions, privacy calculus theory explains the decision-making process in disclosures. These theories are highly relevant to communication and disclosure. First, from the perspectives of privacy calculus and stereotype formation, SCM shows how humans quickly form initial perceptions of others in specific social environments, aiding decision-making based on risks and benefits to meet survival and reproduction needs ([17]). In contrast, privacy calculus theory’s perceived risk and benefit are decision-making tools based on target perception. Similarly to the mechanism behind SCM, the essence of avoiding harm and seeking benefit remains unchanged ([12]). Secondly, in the specific context of information disclosure, privacy calculus theory describes the final step: how individuals decide whether to disclose personal information or not. Frith ([19]) explored decision-making in social contexts, noting that individual decisions depend on understanding others’ intentions, emotions, and beliefs, with stereotypes playing a key role in this process. Comparing perceived risks and benefits requires a basic understanding of the interaction object. Stereotypes serve as a cognitive shortcut when information is insufficient or overwhelming, linking perception and behavior. In conclusion, based on these two theories, a hypothesis was proposed that, in interactions with AI voice assistants, users first form initial perceptions of voice assistants through stereotypes. These perceptions lead users to evaluate potential risks and benefits, ultimately influencing their privacy disclosure decisions. Stereotypes shape how users perceive these intelligent agents, thus affecting their privacy disclosure behaviors.

Researchers have begun to investigate the impact of stereotype formation on information disclosure, suggesting that stereotypes influence individuals’ disclosure processes. [45] ([45]) showed that users infer AI agents’ gender from voice cues and adjust their trust based on gender stereotypes. When users perceive that an assistant’s gender aligns with stereotypical expectations of task ability (e.g., female voice assistants in medical contexts), they are more willing to disclose health-related information. Grimmelikhuijsen ([22]) observed that users use stereotypes to fill in gaps in information, influencing disclosure outcomes. What is more, many studies combine stereotype theory and privacy calculus theory. [8] ([8]) explored how personal resumes on social media influence privacy calculus, showing that people form first impressions based on limited information from social media. The more complete the information, the higher the expected benefits and lower the perceived risks. [21] ([21]) studied how stereotypes about operators lead to misperceptions, affecting privacy risk perception. These studies demonstrate the universal relationship between stereotypes and information disclosure and show that both the stereotype content model and privacy calculus theory are applicable to similar research. However, further exploration is needed to understand how stereotypes influence privacy calculus in specific scenarios. Based on these studies, we proposed the hypothesis that users’ stereotypes of AI voice assistants, mediated by privacy calculus dimensions, will ultimately lead to changes in disclosure intentions:

**H3a/b:** 
*Users’ perceived warmth of voice assistants affects their disclosure intentions through perceived risks/benefits.*


**H4a/b:** 
*Users’ perceived competence of voice assistants affects their disclosure intentions through perceived risks/benefits.*


### 2.3. Stereotype of Voice Gender

Social identity theory tends to view certain social categories or identities as superior to others, leading perceivers to focus on specific categories or identities ([35]). Specifically, categorizing others based on gender and age tends to take priority ([49]). Building on SCM, the role of voice cues in gender perception cannot be overlooked. The design of voice cues in specific contexts has important implications for the design of intelligent voice assistants. By selecting appropriate voice characteristics, it is possible to influence users’ perceptions and attitudes significantly ([39]). Gender traits, which are symbolic markers of social groups, influence users’ trust and preferences with regard to intelligent agents in various contexts. Gender stereotypes suggest that men are often viewed as confident and capable, while women are perceived as warm, detailed, and caring ([67]; [14]).

Research has demonstrated the significant impact of voice gender cues on stereotypes. [45] ([45]) showed that even when users are told they are interacting with a non-human, gender-neutral AI, the voice’s gender characteristics still influence their stereotypes and interaction styles. Male voices are typically linked to competence and trustworthiness, while female voices are more likely to evoke feelings of warmth and emotional resonance. These stereotypes become especially pronounced in certain occupational roles. For instance, in healthcare contexts, women are seen as more empathetic and warmer, while men are often associated with greater technical expertise ([15]). Ernst ([14]) found that users prefer “gendered” robots when the tasks align with gender stereotypes. In his study, users favored female robots for healthcare tasks, while male robots were preferred for security tasks, even though the robots’ abilities were identical. However, gendered designs do not always produce only positive effects in specific contexts. While gendered intelligent agents can enhance users’ feelings of emotional connection, such anthropomorphizing may also provoke discomfort and the uncanny valley effect ([33]). The discomfort arises from psychological classification conflicts and the perceived threat to human uniqueness, which may lead to users’ aversion to these anthropomorphized agents. Based on these studies, we proposed the hypothesis that a voice’s gender, as the independent variable, will influence users’ perceptions of the warmth and competence of the voice assistant:

**H5a/b:** 
*Voice gender affects users’ perceived warmth/the competence of the voice assistant.*


### 2.4. The Self-Disclosure Strategy of AI Affects Stereotypes

Reciprocal self-disclosure is a widely acknowledged method of information acquisition. It involves providing personal information to users, which encourages them to reciprocate by disclosing similar information ([44]). In interactions with users, AI voice assistants play a dual role in advancing the conversation: guiding users to disclose information and also disclosing information about themselves. Mutual self-disclosure in human communication is an integral aspect of social role-playing ([42]; [58]). Individuals use self-disclosure to construct their identity and foster connections with others ([52]). Within the CASA framework, AI is treated (or is expected to be treated) as an integral part of human society, which means disclosures in communication are inherently bidirectional, rather than unidirectional. According to Altman and Taylor’s Social Penetration Theory ([1]), communication tasks require enrichment with personal information. Consequently, AI voice assistants should prepare relevant viewpoints and information. Thus, the AI voice assistant needs to prepare corresponding views on the communication information list. The disclosure strategies of AI voice assistants—both the content and method—play a significant role in shaping users’ perceptions of the assistant’s identity and intimacy, thus influencing the formation of stereotypes ([25]).

[52]’s ([52]) study explored how AI voice assistants employ reciprocal self-disclosure to extract information from users. AI’s self-disclosure can also utilize this reciprocal social model to prompt users to feel compelled to reciprocate by sharing their own information ([55]). However, some studies ([10]) indicate that using reciprocal self-disclosure as an interactive strategy can backfire if users perceive themselves as being manipulated into revealing information. This can result in negative responses, including a loss of trust in the AI assistant, cessation of service use, or providing false information in future disclosures ([10]). [52]’s ([52]) study suggests that reciprocal self-disclosure enhances the anthropomorphism of AI, which strengthens cognitive trust. Herzog applied this self-disclosure strategy to healthcare robots ([25]). The study found that robots using this strategy were more likely to be perceived as anthropomorphic, with higher chances of trust-building. Herzog noted that, compared to other trust-influencing factors, the advantage of self-disclosure is that it does not require significant changes in design, behavior, or underlying technology, demonstrating its practical value and feasibility. Based on these findings, this study explores the self-disclosure strategies of voice assistants as an independent variable and hypothesizes that different types of self-disclosure strategies will influence users’ stereotype perceptions of these assistants.

To categorize self-disclosure strategies, [52]’s ([52]) study adopted Archer and Berg’s ([2]) self-disclosure measurement to assess and manipulate the content of AI voice assistant disclosures in the experiment. Archer and Berg categorized self-disclosure into objective content (e.g., biographical information) which is low in intimacy and subjective content (e.g., fears, self-concept, values) which is high in intimacy. They classified disclosures into four categories, from low to high intimacy: basic information, simple and visible information, attitudes and opinions, and strong affections and basic values ([2]). In postpartum follow-up phone calls, the communication time with users is limited. Unnecessary or excessive information must be avoided to prevent user dissatisfaction. Under the constraint of time, the information disclosed to users typically includes physical condition, psychological well-being, and information related to healthcare services ([78]). These disclosures can be categorized into objective content (e.g., assistant name, processing of information to be provided, postpartum information and knowledge, hospital-related details) and subjective content (e.g., values, empathy, emotional support). These categories align with Archer and Berg’s classification system. The objective disclosures contain basic and visible information, and the subjective disclosures involve attitudes, opinions, and strong emotions. This study adopts Saffarizadeh’s classification method ([52]) to design three self-disclosure strategies for voice assistants, based on levels of intimacy: (1) normal interaction without additional information, (2) intentional disclosure of objective facts, and (3) intentional disclosure of subjective feelings and opinions. The purpose of this study is to explore how these strategies affect user self-disclosure. According to the reciprocity theory of disclosure, the sharing of valuable information is likely to elicit reciprocal disclosures. Disclosure of objective content may enhance perceived competence or increase perceived warmth because of reciprocity, fostering greater trust in the assistant. Additionally, according to Social Penetration Theory ([1]), the depth of interpersonal relationships is determined by the degree and content of self-disclosure. Greater intimacy may lead to closer psychological proximity and influence stereotype formation. For example, subjective disclosures may boost perceived warmth, prompting users to adopt different disclosure strategies. This study proposes the following hypothesis:

**H6a/b:** 
*Different AI self-disclosure strategies affect users’ perceived warmth/competence of the voice assistant.*


Based on these theoretical assumptions, the hypothesis model is constructed, as shown in Figure 1.

The hypothesis model posits that the voice gender of the assistant and its self-disclosure strategies can influence users’ perceived stereotype of the assistant. It hypothesizes that users form a preliminary perception of the assistant based on stereotypes. These initial perceptions lead users to evaluate the potential benefits and risks, which guides their decision to disclose personal information. Stereotypes influence privacy disclosure behaviors by shaping users’ perceptions of these intelligent agents. In the scenario of postpartum follow-up phone calls, perceived risks have a negative effect on disclosure intention (H1). Perceived benefits have a positive effect on disclosure intention (H2). Users’ perceived warmth and competence affect disclosure intention though perceived risks and benefits (H3a/b, H4a/b). Voice gender and self-disclosure strategies affect users’ perceptions of the assistant’s warmth and competence (H5a/b, H6a/b).

## 3. Methods and Procedure

### 3.1. Study Design

A mixed experimental method was used in this study: 2 (voice gender of the telephone follow-up assistant: male voice/female voice) × 3 (information disclosure strategies of the follow-up phone call assistant: normal communication without additional disclosure/intentional disclosure of objective facts/intentional disclosure of subjective feelings and opinions) to explore how different voice genders and disclosure strategies impact users’ intention to self-disclose in AI-driven postpartum follow-up phone calls. Participants were randomly assigned to one of six experimental groups, in which they interacted with the corresponding voice assistant and engaged in the telephone follow-up.

### 3.2. Material Design

#### 3.2.1. Experimental Content Design

This study is based on interviews with hospital staff and relevant postpartum follow-up training literature. The hospital’s postpartum follow-up consists of five main sections: topic introduction, inquiry about the mother and baby’s health status, survey on medical service quality, investigation of medical ethics issues with regard to hospital staff, and finally, responding to user feedback and concluding the follow-up phone call. In the topic introduction phase, the voice assistant greets the participant, introduces itself, and explains the purpose of the call. In the health status inquiry phase, considering common postpartum health issues consists of psychological (e.g., postpartum anxiety) and physiological (e.g., postoperative recovery) aspects ([78]), this study follows relevant postpartum health follow-up guidelines and selects one question related to each aspect to ask the participant ([50]). In the medical care quality survey phase, the voice assistant inquiries about the user’s satisfaction with the medical services and requests suggestions for improvement. In the medical ethics survey phase, the voice assistant encourages the user to disclose any unreasonable experiences they have had during the medical service process and assures the participant that their feedback will be addressed. Finally, the voice assistant responds to the user’s feedback, expresses gratitude, and concludes the follow-up phone call.

In the self-disclosure strategy variable section, the self-disclosure measurement developed by Archer and Berg ([2]) is adopted to evaluate and manipulate the voice assistant’s self-disclosure content. This method classifies disclosed information into four categories: basic information, simple and visible information, attitudes and opinions, and strong affections and basic values, assigning corresponding scores to assess the intimacy of disclosure (see Appendix A, Table A1). In the experiment, based on the first voice follow-up script, which only disclosed basic information, the second script added objective factual disclosures and simple opinion outputs. The third script disclosure revealed more intense emotions and values from the voice assistant’s perspective. The word count across the phases of the three scripts is consistent (seen in Appendix B, Figure A1).

Regarding sound feature control, this experiment examines the impact of voice gender (male/female) on user self-disclosure intention. The average fundamental frequency (F0) for male Mandarin speakers is approximately 120 Hz, and for female speakers, it is around 210 Hz ([60]). To control voice gender, the study employs iFLYTEK’s AI speech synthesis platform, generating audio with standard male (120 Hz) and female (210 Hz) Mandarin voices. The audio rate was 275 words per minute, with pauses strategically placed at the end of each question to allow participants time to think and respond. Each audio clip lasted approximately 1 min and 20 s (±2 s). To ensure participants could recognize the voice content, standard Mandarin pronunciation was used throughout the experiment.

#### 3.2.2. Reciprocal Self-Disclosure Manipulation

To ensure that participants could clearly distinguish between the different disclosure strategies, a pre-test was conducted with three groups of 30 Chinese women. All of the participants had given birth within the past two years. The pre-test was conducted using the Credamo online platform. Participants were randomly assigned to different disclosure content scripts, and the self-disclosure level in each script was rated on a 1–7 scale based on the self-disclosure measurement by [2] ([2]). The results indicated that participants could effectively differentiate between the disclosure strategies used in the scripts (N = 30, M1 = 1.900; M2 = 3.733; M3 = 5.467, F = 90.214, *p* < 0.001). This suggests that the manipulation of disclosure strategies was successful in achieving the intended effect.

#### 3.2.3. Variable Measurement

The measurements for perceived warmth and perceived competence were adapted from the SCM dimensions proposed by [11] ([11]) and the scale applied by [57] ([57]) for voice assistants. To measure perceived risk, perceived benefit, and disclosure intention, the privacy calculus theory measurements were primarily based on the privacy calculation scale used by [75] ([75]) in health communities, the modified version by [65] ([65]), and Wheeless’s self-disclosure scale ([68]). Given the experimental context, the scale was carefully screened, translated, and adjusted to ensure relevance. This process resulted in a final version consisting of 25 items, rated on a 7-point Likert scale, where a higher score indicates stronger agreement, as can be seen in Table 1.

### 3.3. Experiment Process

The study was conducted via the Credamo online platform. We recruited Chinese female participants aged 18–45 who had at least one child. Among them, participants who had given birth within the past two years were selected ([5]; [6]; [23]). Upon starting the experiment, participants were randomly assigned one of six voice materials to listen to. After each segment, they were asked to provide a brief response. After the entire voice material had finished playing and the follow-up phone call concluded, participants filled out the questionnaire provided. The experiment concluded once the questionnaire was completed. Participants who exited the experiment prematurely or failed to listen to the voice material in its entirety were excluded from the data analysis. The experimental process is shown in Figure 2.

The male/female voices here are based on specific voice characteristics derived from SCM and do not represent particular groups. The voice samples are used solely to stimulate the formation of stereotypes. Given that voice characteristics of other genders lack of clear definitions and do not show significant differences, this experiment does not consider voice characteristics of other genders. The data collection for the experiment started in July 2024 and continued until September 2024, with supplementary data collected in October 2024. All participants signed informed consent agreements. The ethics review of the experiment was conducted by the School of Design Art & Media Nanjing University of Science & Technology.

## 4. Result

### 4.1. Descriptive Statistics

A total of 407 valid responses were initially collected for this study. However, 27 respondents were excluded for failing the manipulation check, leaving a final sample size of 385. The participants were women aged 18 to 45, who had at least one child and had given birth within the past two years. Prior to testing the research hypotheses, a reliability analysis was performed on the 385 valid responses. The results indicated that all five variable factors had Cronbach’s α coefficients exceeding the recommended threshold of 0.7, confirming the good reliability of the measurement instruments. KMO = 0.823 (>0.8), and Bartlett’s test of sphericity was statistically significant (*p* < 0.01), demonstrating that the questionnaire possessed robust structural validity. All statistical analyses were conducted using IBM SPSS Statistics 27 software.

### 4.2. The Effect of Voice Assistant Gender and Self-Disclosure Strategies on Perceived Warmth Descriptive

A two-way ANOVA was conducted to explore the effects of voice assistant gender and self-disclosure strategies on users’ perceived warmth, as shown in Figure 3 and Figure 4. The analysis showed no statistically significant interaction between gender and self-disclosure strategies, F(2, 379) = 0.446, *p* = 0.640, indicating that these factors independently influenced perceived warmth. A simple effects analysis revealed that the gender of the voice assistant had a significant effect on perceived warmth. Users reported higher perceived warmth when listening to follow-up phone calls voiced by female assistants (M = 5.783, SD = 0.051) compared to male assistants (M = 5.489, SD = 0.054), F(1, 379) = 15.743, *p* < 0.001, supporting H5a. Self-disclosure strategies also significantly influenced perceived warmth (F(2, 379) = 4.512, *p* = 0.012), providing support for H6a. Specifically, users in the third self-disclosure strategy group (emotional and opinion-based disclosure) reported significantly higher perceived warmth (M = 5.790, SD = 0.062) compared to those in the first group (disclosure as usual) (M = 5.562, SD = 0.065, *p* = 0.039) and the second group (objective based disclosure) (M = 5.555, SD = 0.066, *p* = 0.037). However, no significant difference was observed between the first and second groups (*p* = 1).

### 4.3. The Effect of Voice Assistant Gender and Self-Disclosure Strategies on Perceived Competence

A two-way ANOVA was conducted to explore the effects of voice assistant gender and self-disclosure strategies on users’ perceived competence, as shown in Figure 5. The results indicated no statistically significant interaction between gender and self-disclosure strategies, F(2, 379) = 2.239, *p* = 0.108, supporting H5b. Further analysis of the main effects showed that voice assistant gender had a significant effect on perceived competence. Users perceived higher competence when listening to female-voiced follow-up phone calls (M = 5.896, SD = 0.042) compared to male-voiced calls (M = 5.767, SD = 0.044), F(2, 379) = 4.393, *p* = 0.037. However, self-disclosure strategies did not have a statistically significant effect on perceived competence (*p* = 0.125), leading to the rejection of H6b.

### 4.4. The Direct Effects of Voice Assistant Gender and Self-Disclosure Strategies on Other Variables

A one-way ANOVA was conducted to examine the potential direct effects of voice assistant gender and self-disclosure strategies on users’ self-disclosure behaviors, including perceived benefits, perceived risks, and self-disclosure intention. The analysis revealed no statistically significant direct effects of voice assistant gender or self-disclosure strategies on any of these variables (*p* > 0.05), suggesting that these factors do not independently influence users’ perceptions or intention to disclose information.

### 4.5. Use Structural Equation Modeling (SEM) to Analyze the Relationship Between Stereotypes, Privacy Calculus, and Disclosure Intention

Based on the theoretical model, a structural equation modeling (SEM) path is constructed, as illustrated in Figure 6. Each factor has five variables measured by Likert scales. The study hypothesizes that perceived warmth and perceived competence directly influence perceived risks and perceived benefits. Furthermore, perceived risks and perceived benefits directly influence the user’s disclosure intention. Additionally, the influence of perceived competence and perceived warmth on disclosure intention was also explored.

Calculate the factor loadings for each variable, as shown in Table 2. For the factor warmth, all variables exhibit significant positive loadings, with standardized load factors ranging from 0.593 to 0.693. All variables of factor competence exhibit significant positive loadings, with standardized load factors ranging from 0.547 to 0.606. For the factor risks, all variables exhibit significant positive loadings, with standardized load factors ranging from 0.605 to 0.795. For the factor benefits, all variables exhibit significant positive loadings, with standardized load factors ranging from 0.605 to 0.795. For the factor intention, all variables exhibit significant positive loadings, with standardized loading coefficients ranging from 0.528 to 0.700. The z-values for every variable are far above the critical value, with *p* < 0.01, indicating that all variables meet the factor requirements.

Model regression coefficient of each pathway was calculated, as can be seen in Table 3. The direct effect of perceived warmth on perceived risks (warmth → risks) is not significant (reject H3a) (B = −0.136, SE = 0.167, *p* = 0.222). The direct effect of perceived warmth on perceived benefits (warmth → benefits) is shown to be significantly positive (H3b) (B = 0.550, SE = 0.082, *p* < 0.01). The direct effect of perceived competence on perceived risks (competence → risks) is shown to be significantly negative (H4a) (B = −0.359, SE = 0.226, *p* < 0.01). The direct effect of perceived competence on perceived benefits (competence → benefits) shows a significant positive effect (H4b) (B = 0.403, SE = 0.105, *p* < 0.01). The direct effect of perceived risks on self-disclosure intention (risks → intention) shows a significant negative effect (H1) (B = −0.341, SE = 0.033, *p* < 0.01). The direct effect of perceived benefits on self-disclosure intention (benefits → intention) shows a significant positive effect (H2) (B = 0.817, SE = 0.232, *p* < 0.01). Moreover, the direct effects of perceived warmth and perceived competence on self-disclosure intention (warmth → intention; competence → intention) are not significant (B = −0.208, SE = 0.138, *p* = 0.135; B = 0.138, SE = 0.153, *p* = 0.256). The model demonstrates good fit indices (X^2^/df = 2.463 < 3, GFI = 0.833, RMSEA = 0.0062 < 0.08). The Figure 7 shows the path diagram of the structural equation model (SEM) with coefficients.

## 5. Discussion

### 5.1. Main Findings

This study examined the effects of the gender of AI voice assistants and their self-disclosure strategies on users’ perceived warmth, competence, risks, benefits, and self-disclosure intention in postpartum follow-up phone calls in China. Using two-way ANOVA, the results showed that the gender of the voice assistant significantly influenced perceived warmth and competence. Female voice assistants were rated significantly higher in terms of perceived warmth than their male counterparts, indicating that female voices are more effective at reducing psychological distance during follow-up phone calls. This effect aligns with previous research suggesting that female voices in robots are often associated with greater warmth and emotional resonance ([15]). For perceived competence, female voice assistants also received higher ratings, suggesting that users perceived them as more effective and trustworthy in postpartum follow-up phone calls. This result contrasts with prior findings that associate male voices with competence and trustworthiness. The discrepancy may be attributed to the study’s participants, who were women aged 18 to 45 with childbirth experience—a demographic likely to attribute higher competence to females. Additionally, stereotype content models suggest that perceptions of warmth and competence vary across social contexts. Previous studies have demonstrated that in healthcare settings, users tend to prefer female robots, perceiving them as better suited for these tasks ([14]). In this study, both the scenario of postpartum follow-up phone call and the healthcare context likely reinforced the perception that female assistants are more competent for such roles. The unique status of postpartum women is another factor that influences stereotypes. A study conducted in Hong Kong ([7]) revealed that postpartum women prefer services provided by groups that understand their specific needs. This may help explain why both postpartum follow-up tasks and healthcare-related scenarios lead users to associate female gender with a better performance in such roles in this study.

Regarding self-disclosure strategies, the results indicated that strategies involving emotional and opinion-based disclosures significantly enhanced perceived warmth compared to normal conversation or factual disclosures. High-intimacy self-disclosures fostered closer psychological connections with users, eliciting emotional resonance and enhancing perceptions of empathy, friendliness, and trustworthiness. Furthermore, emotional interactions may have increased the assistant’s perceived anthropomorphism, as suggested by [33] ([33]), who found that anthropomorphism in robots enhances perceived warmth. [52] ([52]) further highlighted that when users perceive AI assistants as human-like, personal self-disclosures enhance intimacy and trust, while factual disclosures have similar effects when the assistants are perceived as machines. This implies that users in this study were more likely to perceive the follow-up assistants as human-like. However, self-disclosure strategies did not significantly influence perceived competence. This result indicates that competence judgments are more strongly associated with the assistant’s gender than with disclosure strategies. [17] ([17]) suggested that, while perceived risk and perceived warmth are discussed as two dimensions of a theory, warmth judgments are the primary judgments, whereas competence judgments require more substantial evidence to change. Users’ initial perceptions of warmth are shaped by context, identity, and voice characteristics, while self-disclosure content primarily plays the role of reinforcing these initial impressions. Since competence judgments are less malleable than warmth perceptions, this explains why self-disclosure strategies significantly affected perceived warmth but not perceived competence.

Although the gender and self-disclosure strategies of voice assistants significantly influenced stereotype formation, they did not directly affect users’ self-disclosure, including perceived risks, benefits and intention. This indicates that while gender and disclosure strategies affect users’ perception, they do not directly determine whether users disclose information. Through structural equation modeling (SEM) analysis, the study found that perceived warmth and perceived competence do not directly influence participants’ self-disclosure intention; instead, their effects are mediated by perceived risks and perceived benefits. Specifically, perceived warmth and perceived competence are positively correlated with perceived benefits, while perceived competence is negatively correlated with perceived risks. However, the impact of perceived warmth on perceived risks was found to be non-significant. Users who perceived the assistant as warm or competent were less likely to perceive risks and more likely to expect benefits.

Notably, the effect of perceived benefits was significantly stronger than that of perceived risks, suggesting that users prioritized benefit evaluations over risks concerns when deciding to disclose information. This finding aligns with Social Exchange Theory, which posits that users are more willing to disclose personal information when they anticipate high benefits (e.g., better services or emotional support), even in the presence of certain risks. In the scenario of postpartum follow-up phone calls in China, the relatively low perceived risks of disclosure may reflect users’ trust in medical institutions and the cooperative nature of doctor–patient relationships. In Chinese culture, doctors are generally associated with positive stereotypes ([56]). A study in China on people’s perceptions of doctors’ stereotypes showed that the public holds a positive stereotype of doctors at a rate of 68.97%. However, 31.03% of respondents stereotyped doctors negatively, citing stress, attitude toward patients, medical competence, and ethical concerns as their reasons for this view; privacy-related issues were not mentioned ([16]). In this experiment, perceived warmth did not significantly affect perceived risks. This could be attributed to the fact that Chinese users are likely to associate the negative stereotypes about doctors (such as attitude toward patients, medical competence, and ethics) with a lack of professional ability. Given the positive stereotype of doctors in China, users are more likely to associate the perceived warmth of the voice with personal benefits, while judging the risk of disclosing information based on the perceived competence of the voice.

On one hand, it is widely acknowledged that China is experiencing a low fertility rate ([59]). In 2022, 9.56 million births were recorded, corresponding to a birth rate of 6.77%. On the other hand, pregnant women in China are proving increasingly willing to pay for higher-quality medical services ([76]). The number of hospital discharges from primary care institutions decreased from 4.795 million in 2009 to 1.473 million in 2021, a decline of 69.3%. The low fertility rate, high single-child rate, and the growing demand for high-quality healthcare have led to behaviors driven by the desire for perceived benefits. Furthermore, with Confucianism as its cultural foundation, China is experiencing a shift toward modernized thinking. Women are often expected to maintain composure for the sake of their children, frequently suppressing their own needs and emotions, which can result in isolation, anxiety, and depression ([77]). The process of adapting to the role of a postpartum mother, coupled with external social pressures and the demands of newborn care, can contribute to significant psychological stress ([40]). This explains why postpartum women are more likely to engage in proactive behaviors to seek perceived benefits, such as social and psychological support.

### 5.2. Significance

#### 5.2.1. Academic Value

This study contributes to the CASA framework by exploring the interactions between SCM and privacy calculus theory—two widely acknowledged theories whose relationship has been seldom explored. By applying SCM and privacy calculus theory to the postpartum follow-up scenario, this study affirms the relevance of these theories in this specialized setting. It provides empirical evidence that these theories interact within specific scenario with AI voice assistants, extending the applicability of the CASA framework.

This study emphasizes the importance of voice assistant gender and self-disclosure strategies in shaping how users interact with the AI technology. In particular, it shows that female voice assistants and emotionally oriented disclosure strategies significantly enhance users’ perception of warmth, which indirectly increases their self-disclosure intention. Specifically, emotionally and opinion-based disclosures improve perceived warmth and indirectly affect self-disclosure intention among target users. The disclosure strategy of AI voice assistants has rarely been examined as a factor influencing stereotype formation, making this study a valuable addition to this field.

Furthermore, the results indicate that users’ disclosure behavior is largely motivated by perceived benefits. It suggests that users are more focused on the potential advantages of interacting with the system rather than on associated risks. This finding underscores the unique, context-dependent influences of social environment and specific interaction settings on self-disclosure intentions with voice assistants. By revealing these contextual characteristics, this study provides empirical support for expanding theories on self-disclosure and privacy calculus. This study helps enhance the understanding of human–machine interaction dynamics and disclosure mechanisms in technology-mediated communication.

#### 5.2.2. Practical Value

From a design standpoint, these findings offer important guidance for optimizing voice assistant functionality. In postpartum follow-up phone calls, both perceived warmth and competence positively influence users’ self-disclosure intention. This suggests that any design strategies that enhance these qualities merit exploration. Specifically, when designing voice assistants for postpartum follow-up, it is recommended to prioritize female voice characteristics and emotional and opinion-based disclosure strategies to increase users’ perceived warmth and perceived benefits.

Additionally, this study reveals that, within the Chinese healthcare context, users are more likely to be motivated by perceived benefits, providing hospitals with insights for refining patient communication approaches. This is a useful finding for healthcare services, implying that emphasizing service benefits may be more effective in encouraging user disclosure than focusing solely on risk reduction. This finding also reflects aspects of Chinese unique doctor–patient relationship dynamics, offering valuable reference points for researching healthcare communication in this specific social context. Finally, this study contributes to characterizing the user profile of medical service recipients in China. In particular, it emphasizes that there are many postpartum users who tend to overlook perceived risks in healthcare interactions. This insight suggests that healthcare propaganda should focus on this group to raise their awareness of the risks when disclosure and protect their interests.

By examining user interactions and preferred communication modes, this study aims to facilitate the adoption of AI voice assistants in healthcare scenarios. Since the interaction context plays a crucial role in shaping user stereotypes, future research could further investigate how gender and emotional expression in voice assistants may be optimized across different scenarios to enhance user engagement and trust in AI technologies.

Finally, although this study was conducted in China, the findings hold significant application value in contexts with different cultural backgrounds. Many developing countries are in the same situation as China. They are undergoing a shift toward modernization and the diversification of medical services, which is influenced by traditional cultural stereotypes ([76]). Therefore, the role of perceived benefits in driving self-disclosure should be emphasized. The diversity of the medical workforce is increasing. According to data from the American Medical Association, 30% of practicing physicians belong to minority groups, and 36% are women ([30]). These findings suggest that stereotypes related to women’s voices in specific scenarios may lead to heightened perceived competence. In low-income countries, the medical resources are insufficient to provide comprehensive healthcare. Follow-up phone calls can be a cost-effective approach to help alleviate this issue. The application of AI also addresses the labor-intensive challenges associated with telephone follow-up ([48]). Further research into the application of voice assistant self-disclosure strategies in these regions could help doctors acquire information more efficiently.

## 6. Conclusions

This study underscores the significance of gender characteristics and self-disclosure strategies in shaping users’ experiences interacting with voice assistants during postpartum follow-up phone calls. The findings reveal that female voice assistants enhance users’ perceived warmth and competence, thereby effectively increasing their self-disclosure intention. Emotional self-disclosure strategies also have a positive effect on perceived warmth to encourage users to disclose. This study highlights the pivotal role of users’ perceived warmth benefits in making self-disclosure decisions.

The study has several limitations. First, the experimental design did not account for scenarios where users refuse to answer or interrupt calls because of self-protection mechanisms, reflecting a broader limitation of the privacy calculus theory. Second, the study primarily focused on postpartum users across China, resulting in a relatively broad target population. Given the potential variations in stereotype formation across different social and cultural contexts, the findings may exhibit regional differences within China and the limited generalizability of global contexts. Moreover, postpartum women’s physiological changes over time may influence the outcomes. Validating these findings requires a larger dataset. Future studies could investigate postpartum women’s medical service experiences from a longitudinal perspective or at different time intervals.

In future research, more specific social contexts need to be investigated and the scenarios need to be better tailored, ensuring the practical applicability of the AI voice assistant. Additionally, attention should be paid to the phenomena of call refusal and interruption. These scenarios deserve further investigation to understand their causes and effects. Future studies should focus on optimizing the design of voice assistants to improve user experience and trust across diverse cultural and healthcare environments. By implementing these strategies, user needs can be better met, and medical services can be further improved and developed.

## Figures and Tables

**Figure 1 behavsci-15-00184-f001:**
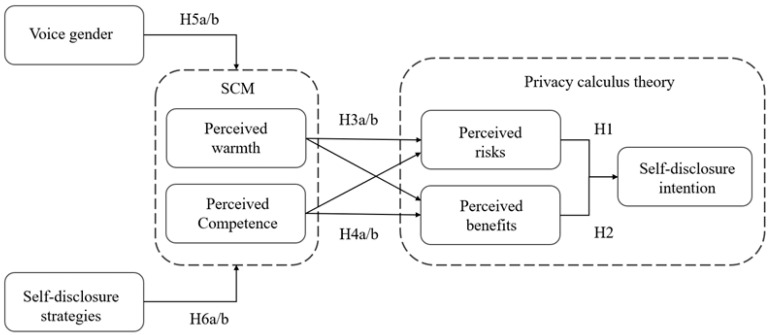
Research hypothesis model.

**Figure 2 behavsci-15-00184-f002:**
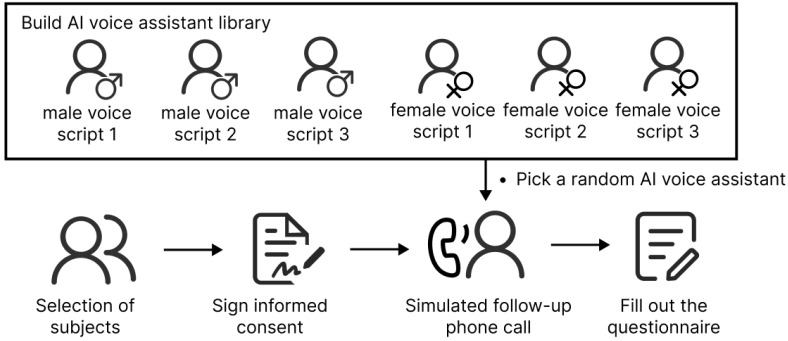
The experimental process.

**Figure 3 behavsci-15-00184-f003:**
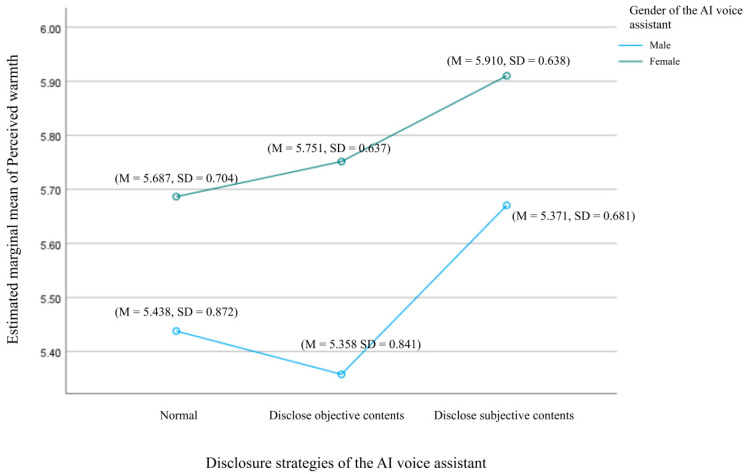
Interaction effects between voice assistant gender and disclosure strategies on users’ perceived warmth: the disclosure strategy is on the X-axis.

**Figure 4 behavsci-15-00184-f004:**
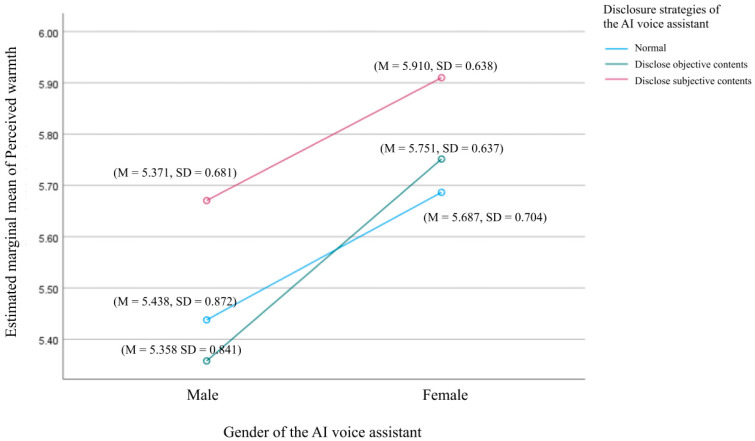
Interaction effects between voice assistant gender and disclosure strategies on users’ perceived warmth: gender is on the X-axis.

**Figure 5 behavsci-15-00184-f005:**
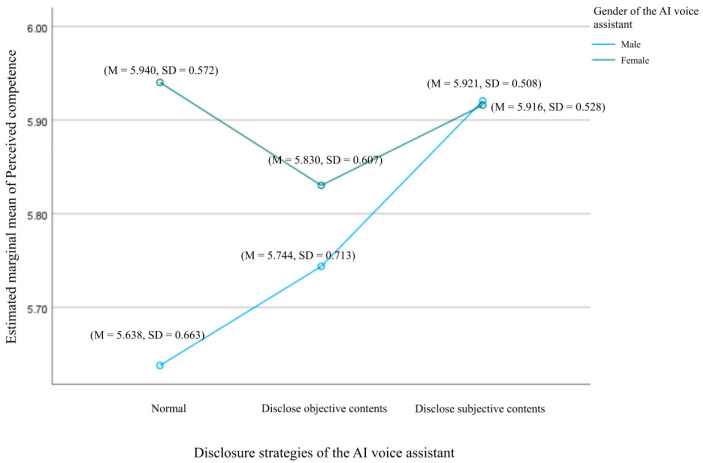
Interaction effects between voice assistant gender and disclosure strategies on users’ perceived competence.

**Figure 6 behavsci-15-00184-f006:**
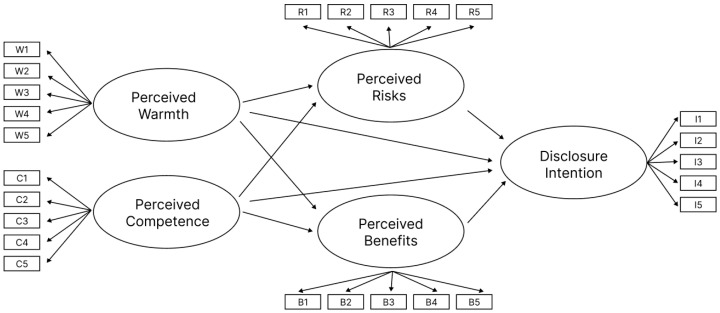
Path diagram of the structural equation model (SEM).

**Figure 7 behavsci-15-00184-f007:**
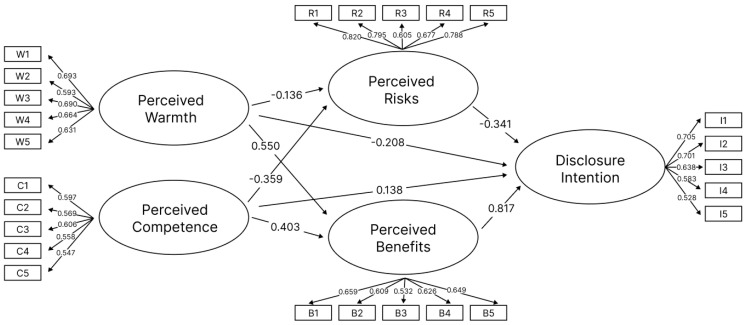
Path diagram of the structural equation model (SEM) with coefficients.

**Table 1 behavsci-15-00184-t001:** Research variables and corresponding measurement items.

Variable Name	Measurement Item	Original Statement	Source
Perceived Warmth	1. I feel the voice assistant in the follow-up phone call is friendly.	1. Friendly	([57]; [11])
	2. I feel the voice assistant in the follow-up phone call is trustworthy	2. Trustworthy	
	3. I feel the voice assistant in the telephone follow-up is good-natured.	3. Good-natured	
	4. I feel the voice assistant in the follow-up phone call is warm.	4. Warm	
	5. I feel the voice assistant in the follow-up phone call is sincere.	5. Sincere	
Perceived Competence	1. I feel the voice assistant in the follow-up phone call is competent.	1. Competent	([57]; [11])
	2. I feel the voice assistant in the follow-up phone call is confident.	2. Confident	
	3. I feel the voice assistant in the follow-up phone call is intelligent.	3. Intelligent	
	4. I feel the voice assistant in the follow-up phone call is efficient.	4. Efficient	
	5. I feel the voice assistant in the follow-up phone call is skilful.	5. Skilful	
Perceived Risks	1. The likelihood of information provided during the follow-up phone call causing risks is high.	1. The likelihood expected problems is high	([38])
	2. Disclosing information during the follow-up phone call has a high potential for loss.	2. The potential for loss would be high	([65])
	3. I believe disclosing private information during the follow-up phone call is inadvisable.	3. I believe that submitting information is not advisable at all	([75])
	4. If my private information is invaded during the follow-up phone call, the consequences would be significant.	4. If my information privacy is invaded, it would be significant	
	5. The follow-up phone call poses a risk of invading my private information.	5. My information privacy is at risk of being invaded	
Perceived Benefits	1. Providing information during the follow-up phone call enables me to receive advice when I need help.	1. Some people would offer suggestions when l needed help	([75]
	2. Providing information during the follow-up phone call helps me overcome challenges when I encounter problems.	2. When l encountered a problem, some people would give me information to help me overcome the problem	
	3. Disclosing information during the follow-up phone call ensures that the follow-up personnel stand by my side when I face difficulties.	3. When faced with difficulties, some people are on my side with me	
	4. Providing information during the follow-up phone call shows the follow-up personnel’s interest and concern for my health and well-being.	4. When faced with difficulties, some people expressed interest and concern in my wellbeing	
	5. Overall, I feel that providing information during the follow-up phone call is beneficial.	5. Overall, I feel that it is beneficial	([65])
Self-Disclosure Intention	1. I intend to provide information during the follow-up phone call.	1. Probable/Not probable	([4]) ([65]; [75])
	2. I am willing to provide information during the follow-up phone call.	2. Willing/Unwilling	
	3. I will continue to disclose information in future follow-up phone calls.	3. Keep reveal	
	4. My self-disclosure during the follow-up phone call accurately reflects my thoughts.	4. When I wish, my self-disclosures are always accurate reflections of who I really am.	([68])
	5. When I disclose information about myself during the follow-up phone call, I do so consciously and intentionally.	5. When I reveal my feelings about myself, I consciously intend to do so.	

**Table 2 behavsci-15-00184-t002:** Standard load factor for each variable.

Factors	Variable	NSlF	SLF	z	SE	*p*
Warmth	w1	1	0.693	-	-	-
	w2	0.754	0.593	10.28	0.073	<0.01
	w3	1.174	0.690	11.758	0.100	<0.01
	w4	1.291	0.664	11.384	0.113	<0.01
	w5	0.861	0.631	10.870	0.079	<0.01
Competence	c1	1	0.597	-	-	-
	c2	1.048	0.569	8.687	0.121	<0.01
	c3	1.147	0.606	9.099	0.126	<0.01
	c4	0.942	0.558	8.562	0.110	<0.01
	c5	0.847	0.547	8.445	0.100	<0.01
Risks	r1	1	0.820	-	-	-
	r2	0.974	0.795	16.791	0.058	<0.01
	r3	0.997	0.605	12.08	0.083	<0.01
	r4	1.049	0.677	13.791	0.076	<0.01
	r5	0.992	0.788	16.616	0.060	<0.01
Benefits	b1	1	0.659	-	-	-
	b2	1.103	0.609	10.424	0.106	<0.01
	b3	1.152	0.532	9.256	0.124	<0.01
	b4	1.163	0.626	10.666	0.109	<0.01
	b5	1.069	0.649	11.004	0.097	<0.01
Intention	i1	1	0.705	-	-	-
	i2	1.017	0.701	12.573	0.081	<0.01
	i3	1.001	0.638	11.506	0.087	<0.01
	i4	0.863	0.583	10.542	0.082	<0.01
	i5	0.827	0.528	9.589	0.086	<0.01

Note: NSLF = non-standard load factor; SLF = standard load factor; z = z-value; and SE = standard error.

**Table 3 behavsci-15-00184-t003:** Model regression coefficient of each pathway.

Effects			B	SE	z	*p*
warmth	→	risks	−0.136	0.167	−1.221	0.222
competence	→	risks	−0.359	0.226	−3.036	<0.01
warmth	→	benefits	0.550	0.082	5.437	<0.01
competence	→	benefits	0.403	0.105	3.983	<0.01
risks	→	intention	−0.341	0.033	−6.740	<0.01
benefits	→	intention	0.817	0.232	4.272	<0.01
warmth	→	intention	−0.208	0.138	−1.493	0.135
competence	→	intention	0.138	0.153	1.136	0.256

Note: B = standardized beta; SE = standard error; z = z-value.

## Data Availability

The raw data supporting the conclusions of this article will be made available by the authors on request.

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
