# Peer review of "Research on the Impact of an AI Voice Assistant’s Gender and Self-Disclosure Strategies on User Self-Disclosure in Chinese Postpartum Follow-Up Phone Calls"

_behavsci, 2025, doi:10.3390/bs15020184_

Round 1
Reviewer 1 Report
Comments and Suggestions for Authors
Dear author(s),
The research presented is relevant in upgrading the existing practice of AI generated postpartum follow-up phone calls although it shows several points that require further elaboration due to its doubtfulness.
The main ethical concern raised in experiment is voluntarily of participants and authors should note if the participants gave an informed consent on the experiment to be conducted voluntarily. Additionally, a critical point of the sample surveyed is a very long postpartum period of two years which needs further elaboration (on why and what grounds was it chosen).
Further on, the reader would also benefit if precise date/time frame of the experiment conducted was noted.
It is also very important to bring in a wider, cross-cultural perspective in theoretical and analysis part in relation to experiences on the protocols from other countries/cultures. Simultaneously, highlighting country (as a case study) could be more visible if noted in Title itself (surely in Abstract/Key words part).
In row 374-375 it is noted how the survey was based on interview with hospital staff. It should be further elaborated what was that specific role of these interviews and how they affected the experiment process itself. It would additionally benefit the overall understanding of the work itself if the originated questionnaire could be appended.
Row 554 please recheck 5th section title – adding Discussion term.
Line 370 notes six experimental groups – please elaborate further, especially in line to 3 different script scenarios.
It is highly likely that not all answers gained fit in straightforward to Table 1 original statements (column 3) if not predetermined by the voice assistant. Please clarify further.
Finally, the methodological and research part is quite robust, but needs further critical literature embeddings to prove its international scientific importance.
Kind regards,
The reviewer
Reviewer 2 Report
Comments and Suggestions for Authors
This paper provides timely insights into improving the quality of healthcare services through the application of AI voice assistants. I would like to encourage authors consider the following points for revision, the research could be further enhanced.
First, asking about postpartum care experiences involves dealing with highly sensitive information. Therefore, it would be necessary to clearly describe the IRB procedures regarding this aspect.
Second, the argument that the gender of the AI assistant affects patients’ perceived experience ultimately relies on gender stereotypes. Wouldn’t it be better to conduct a preliminary survey to determine whether participants hold such gender stereotypes and use this as a basis for discussion? This is because individual perceptions of gender stereotypes may vary significantly.
Third, while the study presents not only the main and interaction effects but also indirect effects through mediators based on the ANOVA and SPSS Process Macro results, it is insufficient for analyzing mediation effects. Structural equation modeling (SEM) would have been more appropriate for deriving accurate results.
Fourth, regarding the references in the introduction, it would be more appropriate to cite foundational studies in the field rather than primarily relying on national, and regional references.
Fifth, while it is intriguing that the hypotheses were presented based on a solid theoretical background, the theoretical background section appears somewhat lengthy.
Sixth, It might be more effective to present the figure illustrating the hypothesis model first, explain each hypothesis in sequence, and conclude with a summary.
Lastly, it would be beneficial to highlight the limitations of this study. Since the analysis methods and findings are closely linked to the Chinese healthcare system, a discussion on the applicability of these results to other cultural contexts could enhance the study's impact. Including the fact that the research was conducted in China in the title and abstract would help provide clarity to readers. Additionally, while the abstract effectively outlines the experimental design, adding a brief mention of the research methodology would further improve its comprehensiveness.
Reviewer 3 Report
Comments and Suggestions for Authors
The article analyzes the impact of gender and strategies of AI-based voice assistants in postpartum follow-up calls. The findings reveal that female voice assistants significantly enhance users’ perceptions of warmth and competence, promoting the disclosure of sensitive information through a positive perception of benefits and minimal risks. Furthermore, emotional and opinion-based disclosure strategies strengthen psychological connection and user trust. These results are valuable for the design of voice assistants in healthcare contexts, helping to optimize interactions and improve service quality. This work can influence the development of more empathetic and effective human-AI interaction technologies, benefiting both patients and healthcare systems.
Below, I would like to offer some comments that could enhance the quality of the manuscript.
General Concept Comments
In general, the article’s content is publishable and does not require substantial structural changes. However, some considerations are presented to enrich and improve the work:
- Abstract: No changes are proposed.
- Introduction: It would be pertinent to include quantitative data on the impact of AI-based voice assistants in assisting women during the postpartum period (lines 35–44). Such information would better contextualize the study’s relevance and support its claims of effectiveness.
- Related Research and Hypothesis:
The theoretical models are well-presented and supported by significant references. Although the hypotheses are grounded in well-established theories, it would be enriching to include quantitative data demonstrating how voice assistants have improved the situation for Chinese women in the postpartum period. Additionally, analyzing how China’s cultural and demographic particularities (e.g., low fertility rates) might influence the results would be beneficial. This would enable a better understanding of the cultural differences in the use of this technology. - Methods and Procedure: The methodology and procedures are adequately and coherently explained.
- Results: The section presents solid evidence on the impact of AI assistant gender and self-disclosure strategies on users’ perceptions and intentions. Moreover, the hypotheses raised are adequately addressed.
- Discussion: The discussion analyzes the results in detail, appropriately relating them to the hypotheses and theoretical framework. However, it could benefit from a gender perspective analysis, exploring how the specific characteristics of the postpartum period might enhance empathy toward female voices. This approach could also consider how motherhood reinforces perceptions of warmth and trust in female voice assistants.
- Conclusions: These are accurate and aligned with the study’s objectives.
Specific Comments
- Include a brief paragraph summarizing the main ideas of Figure 1 (line 360) to facilitate understanding of the proposed hypothesis.
- Improve the presentation of the graphs by adding information about the standard deviation, as this could enhance the understanding of the results
The article provides a solid theoretical analysis of the use of AI-based voice assistants in postpartum follow-up, effectively contributing to improving healthcare systems. However, it could benefit from a deeper discussion of gender and cultural perspectives, considering the specific conditions of Chinese women in the postpartum period. These additions would not only strengthen the interpretation of the results but also broaden the applicability of the study to international contexts.
Round 2
Reviewer 1 Report
Comments and Suggestions for Authors
Dear author(s),
thank you for accepting and implementing the review suggestions into a new paper version. I would like to congratulate you on your performance and thank you for sharing your work on this important and very sensitive topic.
Kind regards,
The Reviewer